# Personalized 3-Gene Panel for Prostate Cancer Target Therapy

**Sanda Iacobas [1]** and **Dumitru Andrei Iacobas [2,\*]**

1    Department of Pathology, New York Medical College, Valhalla, NY 10595, USA; sandaiacobas@gmail.com
2    Personalized Genomics Laboratory, Center for Computational Systems Biology, Roy G. Perry College of Engineering, Prairie View A&M University, Prairie View, TX 77446, USA
\*    Correspondence: daiacobas@pvamu.edu

**Abstract:** Many years and billions spent for research did not yet produce an effective answer to prostate cancer (PCa). Not only each human, but even each cancer nodule in the same tumor, has unique transcriptome topology. The differences go beyond the expression level to the expression control and networking of individual genes. The unrepeatable heterogeneous transcriptomic organization among men makes the quest for universal biomarkers and "fit-for-all" treatments unrealistic. We present a bioinformatics procedure to identify each patient's unique triplet of PCa Gene Master Regulators (GMRs) and predict consequences of their experimental manipulation. The procedure is based on the Genomic Fabric Paradigm (GFP), which characterizes each individual gene by the independent expression level, expression variability and expression coordination with each other gene. GFP can identify the GMRs whose controlled alteration would selectively kill the cancer cells with little consequence on the normal tissue. The method was applied to microarray data on surgically removed prostates from two men with metastatic PCas (each with three distinct cancer nodules), and DU145 and LNCaP PCa cell lines. The applications verified that each PCa case is unique and predicted the consequences of the GMRs' manipulation. The predictions are theoretical and need further experimental validation.

**Keywords:** AP5M1; BAIAP2L1; CRISPR; ENTPD2; master regulator; LOC145474; MTOR; PRRG1; VIM; WFDC3

## 1. Introduction

For decades, cancer genomists have struggled to identify gene biomarkers whose altered sequence (e.g., Reference [1]) or/and expression (e.g., Reference [2]) is/are indicative for the prostate cancer (PCa) and could serve in active surveillance [3] of PCa development. The skillful handling of biomarkers was hoped to increase the survival rate (e.g., Reference [4]), destroy the cancer cells (e.g., Reference [5]) and reduce their proliferation (e.g., Reference [6]) and spreading (e.g., Reference [7]). Biomarkers' "smart" manipulation was also thought to block the PCa recurrence after various types of treatments (e.g., References [8,9]).

However, increasing evidence indicates that most cancerous prostates harbor genetically distinct independently developing malign clones [10]. This tumor heterogeneity [11,12], both at the histopathological and transcriptomic [13,14] levels, within the prostate of one patient, as well as among patients, complicates significantly the diagnostic and treatment options [15–17]. Moreover, together with the biomarker(s) whose altered sequence or expression level is thought indicative for that cancer type, hundreds of other genes are mutated or/and regulated in each cancer nodule with respect to the surrounding cancer-free tissue [18]. The unique combination of affected genes in each human is the direct result of the never-repeatable association of favoring factors affecting the entire body: race, age, medical history, habits, diet, stress, climate, etc. The set of the affected genes depends also on the specific local conditions (microbiome and cellular environment). This explains the observed wide diversity of PCa forms and the large spectrum of treatment

outcomes. Therefore, it is imperious to go beyond precision medicine [12,19] to treatments tailored to the unrepeatable characteristics of every single patient at each moment of his or her life.

This report presents the Gene Master Regulator (GMR) method [20,21] to identify the most legitimate targets for a gene therapy that would selectively kill the cancer cells from the prostate [22]. Although we present three-gene panels to erase three distinct cancer clones in the profiled prostate, the method could be used for as many as relevant cancer nodules are found in the tissue. The GMR of a cell phenotype in a tissue is the gene whose strictly controlled expression level regulates the major functional pathways by coordinating the expressions of most of their genes. Owing to the uniqueness of the transcriptome, each cell phenotype of the tumor has a distinct gene hierarchy. Therefore, the GMR approach personalizes the gene treatment for each patient to destroy as many as possible cancer nodules of his affected prostate.

The GMR method is based on the Genomic Fabric Paradigm (GFP) [23] that takes advantage of profiling thousands of genes at a time on multiple biological replicates. GFP assigns to each quantified gene the independent variables: average expression level (AVE), relative expression variability (REV) and correlation (COR) with expression of each other gene [24]. Regardless of (microarray or RNA-sequencing) platform, adding REVs and CORs values increases by four orders of magnitude the amount of useful information provided by the transcriptomic study.

AVE is used by all oncogenomists to determine whether that gene was up/downregulated or turned on/off by cancer with respect to the normal tissue. In almost all publications, AVE is the single variable considered for individual genes.

Although profiling additional biological replicas was required initially only for statistical relevance of the results, it also gives us very important clues about the cell priorities in controlling the random fluctuations of the gene expression. The biological replicas can be formally considered as instances of the same system subjected to (non-significantly regulating) different environmental conditions. Thus, REV indicates how sensitive that gene is to slight environmental changes beyond the inherent stochastic nature of the chemical reactions involved in the transcription. In all transcriptomic studies, we found genes that are very stably expressed (low REV) and genes with high expression variability (high REV) across biological replicas. Low REV indicates strong control of the expression level by cellular the homeostatic mechanisms, most likely because the right expression of that gene is critical for the cell phenotypic expression, survival, proliferation and integration in the multicellular tissue. By contrast, expressions of other genes are left to fluctuate (high REV) to ensure cell adaptation to the environmental continuous changes [25].

The profiling expressions of thousands of genes at a time on biological replicas allows us to quantify how many fluctuations in the expression of one gene are correlated/coordinated with fluctuations of each other gene across biological replicas. COR analysis responds to the "Principle of Transcriptomic Stoichiometry" [25], a generalization of the Law of Multiple Proportions from chemistry [26]. The principle requires coordinated expression of genes whose encoded products are linked in functional pathways.

## 2. Materials and Methods

### 2.1. Prostate Tissues and Cell Lines

This report uses transcriptomic profiles generated in the NYMC IacobasLab by profiling the surgically removed prostates of two men, hereafter denoted as patients "ABCN" and "PQMZ". For comparison, we added the expression data from two human prostate cancer cell lines: the androgen-sensitive LNCaP [27] and the not-hormone-sensitive DU145 [28]. Expression data obtained in our lab from the LNCaP cells (hereafter denoted by "L") were deposited at Reference [29], and those from the DU145 cells (hereafter denoted by "D") were deposited at Reference [30].

From the "ABCN" patient, we profiled the primary cancer nodule "A" (Gleason Score GS = 4 + 5 = 9); the secondary cancer nodules "B" and "C", each with GS = (4 + 4 = 8);

and the surrounding normal tissue "N". Gene-expression data of "ABCN" were deposited at Reference [31] for the primary nodule "A" and the cancer-free margins "N", and at Reference [32] for the secondary nodules "B" and "C", and partially analyzed in a recent paper [33]. Patient "PQMZ" had prostatic adenocarcinoma involving 75% of bilateral lobes, with extensive perineural invasion, multifoci of extraprostatic extension that affected also the bilateral seminal vesicles. Gene-expression data from this patient are available at Reference [34] for the nodule "M" (GS = 4 + 5 = 9) and the cancer-free resection margins "Z", and at Reference [35] for the cancer nodules "P" and "Q", each with GS = 4 + 5 = 9. From each nodule, we collected a ~2 mm area from the center and then split it into 4 parts to limit the possibility that the collected quarters contain cells from different clones.

The study, conducted according to the guidelines of the Declaration of Helsinki, was part of Dr. Iacobas's project approved by the Institutional Review Boards (IRBs) of the New York Medical College (NYMC) and Westchester Medical Center (WMC) Committees for Protection of Human Subjects. The approved IRB (L11,376 from 2 October 2015) granted access to frozen cancer specimens from the WMC Pathology Archives and depersonalized pathology reports, waiving the patients' informed consent.

The experimental protocol (RNA extraction, fluorescent labeling, hybridization with the microarray and washing and scanning the chip), as well as the primary analysis of the fluorescent values (filtering, background subtraction and normalization to the median of valid spots in all profiled samples) are detailed on the Gene Expression Omnibus website hosting the deposited datasets [29–32,34,35].

### 2.2. Transcriptomic Characteristics of Individual Genes

AVE, REV and COR values were computed to account for the non-uniform numbers of spots probing redundantly numerous genes in Agilent (Agilent, Santa Clara, CA, U.S.A.) 4 × 44 k human dual-color microarrays (configuration G2519F, platform GPL13497 [36]).

$$AVE_i^{(sample)} = \frac{1}{R_i}\sum_{k=1}^{R_i} \mu_{i,k}^{(sample)} = \frac{1}{R_i}\sum_{k=1}^{R_i}\left(\frac{1}{4}\sum_{j=1}^{4} a_{i,k,j}^{(sample)}\right), \quad where:$$

$sample = "N", "A", "B", "C", "Z", "P", "Q", "M", "L", "D"$

$R_i =$ number of spots probing redundantly gene "i",

$a_{i,k,j}^{(sample)} =$ expression of gene "i" probed by spot "k" on biological replica "j" in "sample" (1)

$$REV_i^{(sample)} = \frac{1}{2}\underbrace{\left(\sqrt{\frac{r_i}{\chi^2(r_i;0.975)}} + \sqrt{\frac{r_i}{\chi^2(r_i;0.025)}}\right)}_{\text{correction coefficient}} \sqrt{\underbrace{\frac{1}{R_i}\sum_{k=1}^{R_i}\left(\frac{s_{ik}^{(sample)}}{\mu_{ik}^{(sample)}}\right)^2}_{\text{pooled CV}}} \times 100\%$$

(2)

$\chi^2(r_i;\alpha) =$ chi-square for $r_i (= 4R_i - 1 =$ number of degrees of freedom) and probability $\alpha$

$\mu_{ik} =$ average expression of gene i probed by spot k $(= 1, \ldots, R_i)$ in the 4 biological replicas

$s_{ik} =$ standard deviation of the expression level of gene i probed by spot k

The correction coefficient is the mid chi-square interval estimate of the unit standard deviation and takes values from 2.15 for genes probed by one spot each to 1.05 for genes probed by 11 spots (e.g., *TP53*).

$$COR_{ig}^{(sample)} = \frac{\sum_{k_i=1}^{R_i}\sum_{k_g=1}^{R_g}\left(\sum_{j=1}^{4}\left(a_{i,k,j}^{(sample)} - AVE_i^{(sample)}\right)\left(a_{g,k,j}^{(sample)} - AVE_g^{(sample)}\right)\right)}{\sqrt{\sum_{k_i=1}^{R_i}\left(\sum_{j=1}^{4}\left(a_{i,k,j}^{(sample)} - AVE_i^{(sample)}\right)^2\right)\sum_{k_g=1}^{R_g}\left(\sum_{j=1}^{4}\left(a_{g,k,j}^{(sample)} - AVE_g^{(sample)}\right)^2\right)}}$$

(3)

In Equation (3), *COR* ($-1 \leq COR \leq 1$) is the Pearson product-momentum correlation coefficient between the (log$_2$) expression levels of genes "*i*" and "*g*". For genes probed by 1 spot each, $p < 0.05$ significant synergistic/antagonistic expression was assigned if

$|COR| \geq 0.95$. If the genes are probed by 2 spots each, then significant coordination occurs for $|COR| \geq 0.71$, and so on with cutoff diminishing for larger numbers of spots probing each gene [25].

*REV* and *COR* were used to determine the Gene Commanding Height (*GCH*) [33], which establishes the gene hierarchy in the profiled phenotype:

$$GCH_i^{(sample)} = \underbrace{\frac{\langle REV \rangle^{(sample)}}{REV_i^{(sample)}}}_{\text{estimate of the transcription control}} \times \underbrace{\exp\left(4 \overline{\left(COR_{ig}^{(sample)}\right)^2}\Big|_{\forall g \neq i}\right)}_{\text{measure of expression coordination}}, \quad where:$$

(4)

$\langle \ \rangle =$ median, $\overline{(\ )^2} =$ average of the square values

The top gene of the hierarchy (highest *GCH*) is the Gene Master Regulators (*GMRs*) of that phenotype, whose altered expression should have the largest consequences. The hierarchies of the four groups of samples were used to identify the top 3 genes whose *GCH* scores in the three cancer nodules are far above the corresponding scores in the cancer-free tissue.

### 2.3. Transcriptomic Alterations in Cancer

A gene was considered as statistically ($p < 0.05$) significantly regulated in a cancer nodule ("cancer") with respect to the normal tissue from the same tumor if the absolute fold-change $x$ and the $p$-value ($p_i^{(normal \rightarrow cancer)}$) of the heteroscedastic $t$-test of the means equality in the two regions satisfied the composite criterion Equation (5). Our cutoff of the absolute fold-change considers the combined effects of the biological variability and technical noise, providing a much accurate criterion for expression regulation than any other arbitrary ($1.5\times$, $2.0$) fold-change requirement.

$$\begin{cases} \left| x_i^{(normal \rightarrow cancer)} \right| > CUT_i^{(normal \rightarrow cancer)} = 1 + \frac{1}{100} \sqrt{2\left(\left(REV_i^{(normal)}\right)^2 + \left(REV_i^{(cancer)}\right)^2\right)} \\ p_i^{(normal \rightarrow cancer)} < 0.05 \end{cases}$$

$where: cancer = "A", "B", "C", "M", "P", "Q" \quad normal = "N", "Z"$

(5)

$$x_i^{(normal \rightarrow cancer)} \equiv \begin{cases} \dfrac{\mu_i^{(cancer)}}{\mu_i^{(normal)}} & , \quad \text{if } \mu_i^{(cancer)} > \mu_i^{(normal)} \\ -\dfrac{\mu_i^{(normal)}}{\mu_i^{(cancer)}} & , \quad \text{if } \mu_i^{(cancer)} < \mu_i^{(normal)} \end{cases}$$

The $p$-value was computed with Bonferroni correction for multiple testing [37] in the case of several spots probing redundantly the same gene.

Instead of the uniform $\pm 1$ contribution to the altered transcriptome used in the traditional percentage measure of the significantly up/downregulated genes, we considered the Weighted Individual Gene Regulation (*WIR*). *WIR* analysis is not limited to the significantly regulated genes; it is applied to any gene. *WIR* was used to compute the Weighted Pathway Regulation (*WPR*) to average the contributions of all genes assigned to that functional pathway:

$$WIR_i^{(normal \rightarrow cancer)} = AVE_i^{(normal)} \frac{x_i^{(normal \rightarrow cancer)}}{\left| x_i^{(normal \rightarrow cancer)} \right|} \left(\left| x_i^{(normal \rightarrow cancer)} \right| - 1\right)\left(1 - p_i^{(normal \rightarrow cancer)}\right)$$

(6)

$$WPR_\Gamma^{(normal \rightarrow cancer)} = \overline{WIR_i^{(normal \rightarrow cancer)}}\Big|_{i \in \Gamma}, \quad \Gamma = \text{functional pathaway}$$

## 3. Results

### 3.1. Overview

In total, we quantified the expression of 14,203 distinct unigenes in all three cancer nodules and the surrounding cancer-free tissue from the surgically removed prostate of the "PQMZ" man. The 403,620,854 (total number of AVE, REV and COR) values resulted

from this experiment were used to illustrate the analyses below. In order to show the uniqueness of the three-gene target panel, we used also the expression values in the three cancer nodules and cancer-free surroundings of the "ABCN" man (14,908 genes), in the "L" cells (15,278 genes) and in the "D" cells (16,126 genes).

Table 1 presents the three genes with the largest expression levels in each of the four profiled regions of patient "PQMZ" and in the four regions of the patient "ABCN". Remarkably, *RPL13* was among the three genes with the largest expressions in all regions, and in both patients, it was downregulated in all cancer nodules with respect to the corresponding normal tissues. The robust downregulation of *RPL13* in all six cancer nodules (by $-1.70\times/-1.38\times/-1.36\times$ in "P"/"Q"/"M" vs. "Z" and by $-2.15\times/-1.30\times/-1.50\times$ in "A"/"B"/"C" vs. "N") explains the reduced immune response in cancer. Our results add new evidence about the extra-ribosomal roles of *RPL13*, particularly in activating the innate response [38].

**Table 1.** Three genes with the largest average expression levels (AVE) in each of the four profiled regions of the "PQMZ" patient and in the four regions of the "ABCN" patient. The largest 3 AVE values in each phenotype have a gray background.

| Gene | Description | AVE-P | AVE-Q | AVE-M | AVE-Z |
|---|---|---|---|---|---|
| CYTB | mitochondrially encoded cytochrome b | 231 | 237 | 186 | 266 |
| **RPL13** | **ribosomal protein L13** | 215 | 2365 | 269 | 366 |
| ACTG2 | actin, gamma 2, smooth muscle, enteric | 204 | 126 | 119 | 325 |
| NPY | neuropeptide Y | 76 | 471 | 10 | 123 |
| **RPL13** | **ribosomal protein L13** | 215 | 265 | 269 | 366 |
| RPL7A | ribosomal protein L7a | 106 | 310 | 141 | 241 |
| **RPL13** | **ribosomal protein L13** | 215 | 265 | 269 | 366 |
| ZNF865 | zinc finger protein 865 | 182 | 267 | 203 | 292 |
| PQLC2 | PQ loop repeat containing 2 | 172 | 179 | 199 | 203 |
| **RPL13** | **ribosomal protein L13** | 215 | 265 | 269 | 366 |
| ACTG2 | actin, gamma 2, smooth muscle, enteric | 204 | 126 | 119 | 325 |
| MYH11 | myosin, heavy chain 11, smooth muscle | 137 | 101 | 139 | 307 |

| Gene | Description | AVE-A | AVE-B | AVE-C | AVE-N |
|---|---|---|---|---|---|
| **RPL13** | **ribosomal protein L13** | 288 | 477 | 415 | 621 |
| PQLC2 | PQ loop repeat containing 2 | 227 | 266 | 338 | 490 |
| CYTB | mitochondrially encoded cytochrome b | 215 | 173 | 146 | 382 |
| **RPL13** | **ribosomal protein L13** | 288 | 477 | 415 | 621 |
| RPS27 | ribosomal protein S27 | 158 | 365 | 294 | 212 |
| RPL32 | ribosomal protein L32 | 164 | 315 | 299 | 359 |
| **RPL13** | **ribosomal protein L13** | 288 | 477 | 415 | 621 |
| ZNF865 | zinc finger protein 865 | 132 | 215 | 395 | 600 |
| RPS8 | ribosomal protein S8 | 153 | 233 | 348 | 504 |
| **RPL13** | **ribosomal protein L13** | 288 | 477 | 415 | 621 |
| ZNF865 | zinc finger protein 865 | 132 | 215 | 395 | 600 |
| RPS2 | ribosomal protein S2 | 164 | 260 | 337 | 517 |

Table 2 lists the three genes with the most controlled expression (low REV) in each of the four profiled regions of the patient "PQMZ" and in the four regions of the patient "ABCN". Supplementary Table S1 presents the three genes with the most variable expression (high REV) across biological replicas in all profiled regions of the two patients.

**Table 2.** Three most stably (low REV, darker gray background) expressed genes in each of the four profiled regions of patients "PQMZ" and "ABCN".

| Gene | Description | REV-P | REV-Q | REV-M | REV-Z |
|---|---|---|---|---|---|
| FKBP9 | FK506 binding protein 9 | 0.32 | 8.80 | 10.87 | 9.27 |
| ZBTB2 | zinc finger and BTB domain containing 2 | 0.50 | 12.39 | 5.03 | 16.51 |
| NUBPL | nucleotide binding protein-like | 0.69 | 12.19 | 5.48 | 10.73 |
| TBRG4 | transforming growth factor beta regulator 4 | 11.07 | 0.59 | 1.56 | 8.06 |
| DNAJC24 | DnaJ (Hsp40) homolog, subfamily C, member 24 | 11.62 | 0.85 | 7.19 | 4.16 |
| UBE3B | ubiquitin protein ligase E3B | 5.32 | 1.00 | 6.51 | 9.46 |
| TMEM186 | transmembrane protein 186 | 9.32 | 20.45 | 0.28 | 12.49 |
| NDUFA6-AS1 | NDUFA6 antisense RNA 1 (head to head) | 28.15 | 8.46 | 0.50 | 15.82 |
| LMAN2L | lectin, mannose-binding 2-like | 8.05 | 8.20 | 0.52 | 8.00 |
| COPS5 | COP9 signalosome subunit 5 | 51.46 | 27.83 | 7.37 | 0.12 |
| ARPC5L | actin related protein 2/3 complex, subunit 5-like | 19.90 | 22.18 | 7.13 | 0.14 |
| DAZAP1 | DAZ associated protein 1 | 7.14 | 18.87 | 3.84 | 0.22 |
| **Gene** | **Description** | **REV-A** | **REV-B** | **REV-C** | **REV-N** |
| ENTPD2 | ectonucleoside triphosphate diphosphohydrolase 2 | 0.29 | 33.36 | 15.97 | 8.66 |
| COMMD9 | COMM domain containing 9 | 1.04 | 23.01 | 4.49 | 3.82 |
| MIEN1 | migration and invasion enhancer 1 | 1.14 | 13.78 | 9.01 | 8.67 |
| SSX3 | synovial sarcoma, X breakpoint 3 | 39.80 | 0.94 | 17.16 | 12.18 |
| FCRL5 | Fc receptor-like 5 | 121.40 | 1.41 | 36.36 | 38.96 |
| RANBP2 | RAN binding protein 2 | 48.44 | 1.63 | 11.21 | 10.48 |
| BAIAP2L1 | BAI1-associated protein 2-like 1 | 64.94 | 11.35 | 0.40 | 8.03 |
| FAM71E1 | family with sequence similarity 71, member E1 | 41.61 | 48.01 | 0.81 | 11.93 |
| LILRB3 | leukocyte immunoglobulin-like receptor, subfamily B, member 3 | 22.77 | 32.51 | 0.91 | 27.55 |
| MRPS12 | mitochondrial ribosomal protein S12 | 34.96 | 42.93 | 11.21 | 0.32 |
| TOR1A | torsin family 1, member A | 11.09 | 17.64 | 6.28 | 0.42 |
| DENND1B | DENN/MADD domain containing 1B | 8.18 | 45.86 | 15.79 | 0.47 |

Table 2 has some very interesting results. First, each of the four regions from each of the two patients appears to have different priorities in controlling the transcripts' abundances. Our results indicate that the most controlled genes are critical for preserving the phenotype. Thus, *FKBP9*, the most controlled gene in nodule "P", is known for promoting malignant behavior of glioblastoma cells [39] and poor prognosis of PCa patients [40]. *TBRG4*, the most stably expressed gene in "Q", was reported as being actively involved in myeloma [41], squamous carcinoma [42], osteosarcoma [43], glioblastoma [44], leukemia [45] and lung cancer [46]. The list of stably expressed genes also includes a long noncoding RNA, *NDUFA6-AS1*, identified recently as a biomarker for the prognostic of thyroid cancer [47]. We believe that the strict control of the expression of *COPS5* is related to its role in controlling the progression of PCa [48].

Supplementary Table S1, which presents the most variably expressed genes across biological replicas, is also interesting by indicating that the adaptation to the environmental fluctuations is carried out by distinct sets of genes in each profiled region. Thus, excepting *UBE2I*, common for nodule "M" and normal tissue "Z", there is no overlap of the most variably expressed three genes in profiled regions.

### 3.2. Independent Variables

Figure 1 serves as an example of the independency of AVE, REV and COR for the first 50 alphabetically ordered genes involved in the mTOR signaling [49] in the cancer nodules "P", "Q", and "M" and the normal surrounding tissue "Z" of the "PQMZ" patient prostate. Figure 1c presents the expression coordination with *MTOR* (mechanistic target of rapamycin (serine/threonine kinase)).

Although the *MTOR* gene and its partners in the mTOR signaling pathway were selected for their roles in the development, proliferation and migration of cancer cells [50], any other subset of genes would confirm the independence of the AVE, REV and COR characteristics.

Within this selection, genes such as ATPase, H+ transporting, lysosomal 13 kDa and V1 subunit G2 (*ATP6V1G2*) have very low expression (AVE = 0.14 in "Q"), and genes such as ATPase, H+ transporting, lysosomal 14 kDa and V1 subunit F (*ATP6V1F*) have much higher expression (34.5 in "Q"). Likewise, there are very stably (e.g., DEP domain containing five (*DEPDC5*), REV = 0.25% in "Z") and very unstably (e.g., frizzled class receptor 10 (*FZD10*), REV = 87.48% in "Q") expressed genes.

In addition to the clear independence of the three variables, of note are also the differences among the three equally histopathologically ranked cancer nodules from the same prostate. These findings prove both the transcriptomic heterogeneity of the PCa and the uniqueness of each nodule. The transcriptomes of the three nodules differ not only by expression levels of individual genes but also in their expression variability (indicating different strengths of controlling mechanisms) and expression coordination (distinct gene networking in pathways). For instance, *FZD10*, known for its role in breast cancer [51] was very unstably expressed in "Q" and "P" (REV = 59.96%), but it was stably expressed enough in "M" (REV = 6.16%). These values suggest that the right expression of *FZD10* was more important for the "M" cells than for the "P" and "Q" cells. Moreover, eukaryotic translation initiation factor 4E (*EIF4E*) was synergistically expressed with *MTOR* in "Q", but antagonistically expressed with *MTOR* in "P" and "M", meaning that *EIF4E* might work as an activator of *MTOR* in "Q" but as an inhibitor in "P" and "M", which indicates that targeting *EIF4E* may have opposite clinical results [52]. Warning: the "opposite roles", suggested by a pure theoretical speculation about in-phase and in-antiphase fluctuations of the expressions of *MTOR* and *EIF4E* in distinct nodules, needs rigorous experimental validation.

*3.3. The Power of the Weighted Individual (Gene) Regulation (WIR) Score to Identify the Main Contributors to the Cancer Phenotype*

Figure 2 presents the regulation of the first 50 alphabetically ordered mTOR signaling genes in the three cancer nodules with respect to the cancer-free surrounding tissue. The regulation of genes is presented as uniform $\pm 1$ contribution (as in the percentage of significantly up/downregulated genes), expression ratio "x" and Weighted Individual Gene Regulation (WIR).

The power of WIR to discriminate the genes according to their contribution is outstanding. For instance, *DEPTOR* (DEP domain containing MTOR-interacting protein) and *DVL1* (disheveled segment polarity protein 1) have similar expression ratios in "Q" (2.03× and 2.09×) but substantially different WIRs (1.10 and 17.13).

Remarkably, although most of the significant regulations go the same way in all three nodules (e.g., *ATP6V1AB2*, *ATP6V1C1*, *ATV6V1H*, *FLCN*, *FZD1* and *FZD3*), suggesting a shared PCa transcriptomic signature. However, none of the similarly regulated genes was listed among the PCa biomarkers in the NIH-NCI GDC Data Portal [53], and there are also opposite regulations (*AKT1*, *DVL2* and *DVL3*). The significant opposite regulations presented in Table 3 and Figure 2 indicate that, even within the same tumor, each cancer nodule has its own cancer transcriptomic topology. Of note is that almost all human genes (20,237) were found to be altered in at least one case of PCa included in the portal [53], and the top PCa biomarkers are among the top biomarkers for many other cancers. For instance, *TP53* (#1 for PCa) is also #1 for lung, breast, head and neck, ovary cancers and among the top five for almost all other cancers. Therefore, we believe that it makes no sense to continue using the transcriptomic signature in classifying the PCas and that the commercially available assays for cancer diagnostic (e.g., References [54–56]) have disputable prediction value, as discussed in a previous paper [33].

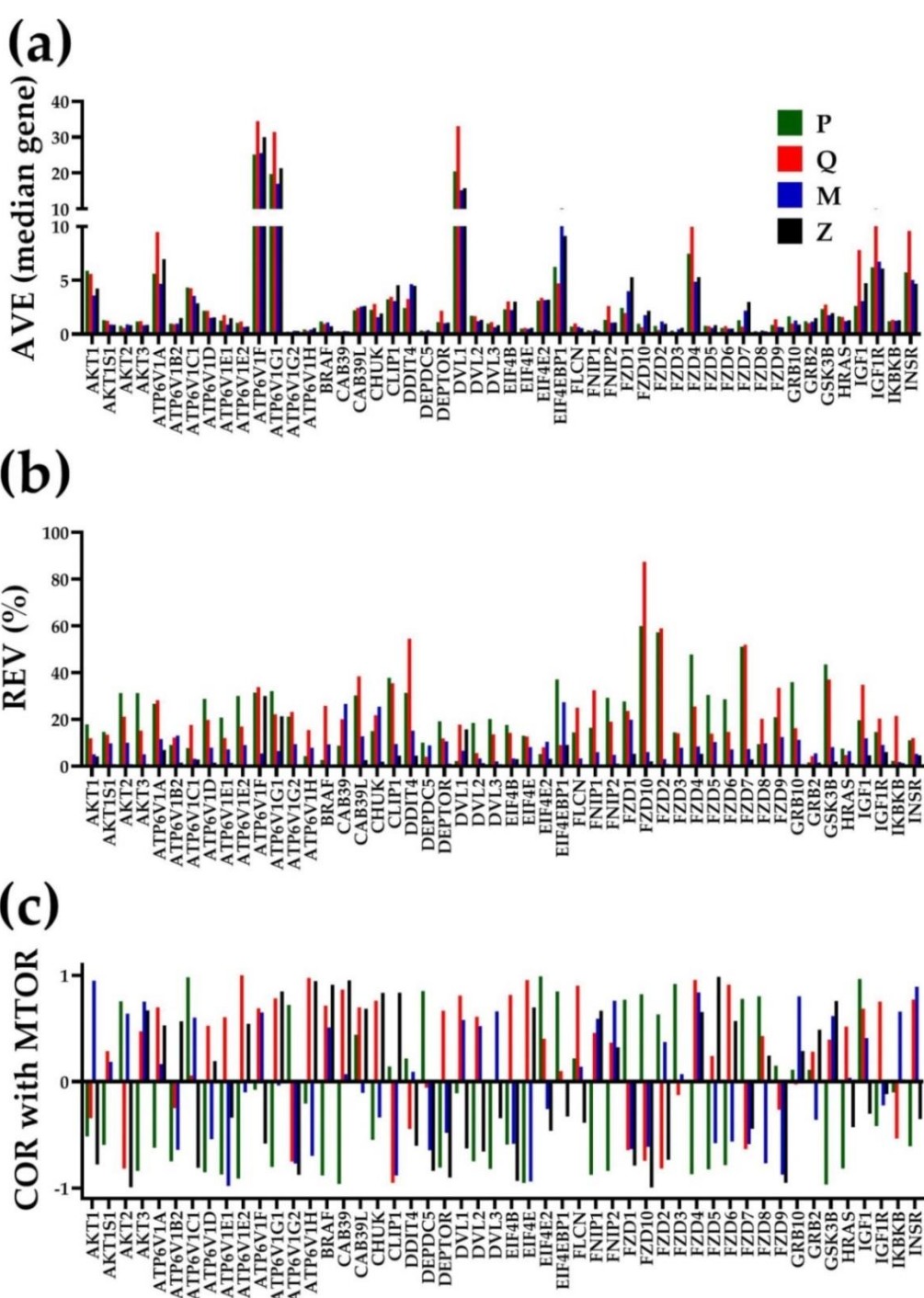

**Figure 1.** Independent transcriptomic characteristics of the first 50 alphabetically ordered genes of the mTOR signaling pathway in the three cancer nodules ("P", "Q" and "M") and the surrounding normal prostate tissue ("Z"). (**a**) Average expression levels (AVE) in expressions of the median gene. (**b**) Relative expression variability (REV). (**c**) Expression correlation (COR) with *MTOR*.

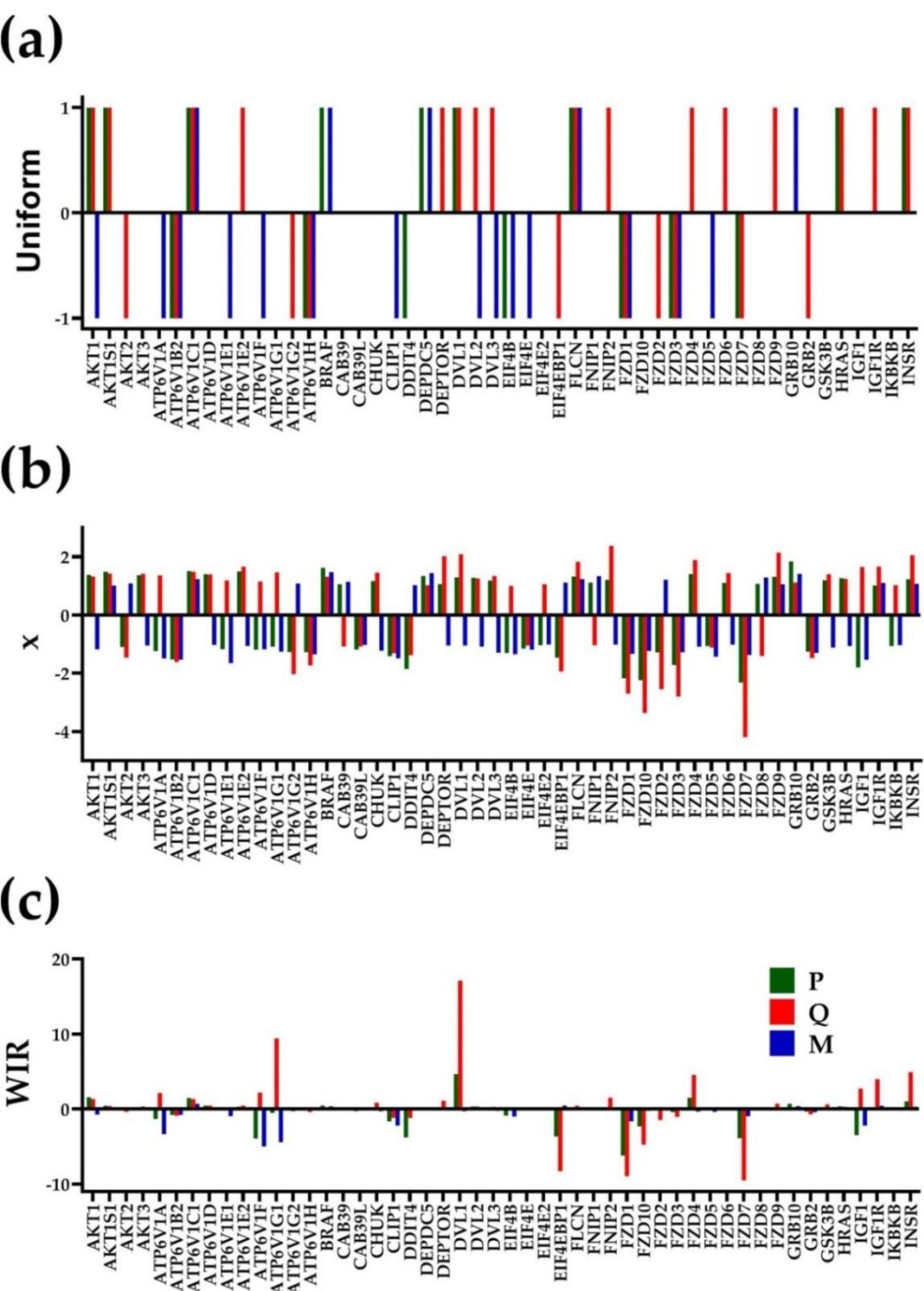

**Figure 2.** Regulation of the first 50 alphabetically ordered mTOR signaling genes. (**a**) As uniform $\pm1$ contribution. (**b**) Expression ratio "x" (negative for downregulation). (**c**) Weighted Individual Gene Regulation (WIR). Note the differences among the nodules.

### 3.4. Each Cancer Nodule Has Its Own "Transcriptomic Signature"

Table 3 presents the three genes with the largest positive and the three genes with the largest negative contributions to the transcriptomic alterations in each of the three cancer nodules of the patients "PQMZ" and "ABCN".

It is interesting to note that, in Table 3, each nodule has different sets of the three largest contributors and that these contributors are involved in a wide diversity of functional pathways. These data indicate a large spectrum of possible molecular mechanisms responsible for the formation of cancer nodules in the prostate. Moreover, while some

genes were regulated the same way in all three nodules (*CNN1*, *RNA28S5*, *RLN1*, and *ACTG2*), others (*RPS8*, *MARC1*, and *PSCA*) were regulated in only one or two nodules. There are also genes (e.g., *NPY*, *IGKC*, *IGHG1*, and *SNORD3B-1*) that were even oppositely regulated in one nodule than in the other two. These results indicate the unique response of each region to cancer and that restoration of the normal expression level of some genes might have opposite effects on distinct nodules.

**Table 3.** Three genes with the largest positive and negative contributions to the transcriptomic alterations in the cancer nodules of the "PQMZ" and "ABCN" patients.

| Gene | Description | WIR-P | WIR-Q | WIR-M |
|------|-------------|-------|-------|-------|
| PSCA | prostate stem cell antigen | **54** | 138 | 0 |
| KLK12 | kallikrein-related peptidase 12 | **40** | 121 | 0 |
| BASP1 | brain abundant, membrane attached signal protein 1 | **32** | 39 | 0 |
| RPS8 | ribosomal protein S8 | **−383** | 0 | −123 |
| CNN1 | calponin 1, basic, smooth muscle | **−403** | −680 | −568 |
| RNA28S5 | RNA, 28S ribosomal 5 | **−604** | −314 | −26 |
| MARC1 | mitochondrial amidoxime reducing component 1 | 0 | **9180** | 0 |
| NPY | neuropeptide Y | −50 | **348** | −1345 |
| PSCA | prostate stem cell antigen | 54 | **138** | 0 |
| CNN1 | calponin 1, basic, smooth muscle | −403 | **−680** | −568 |
| LTF | lactotransferrin | −301 | **−1682** | 22 |
| RLN1 | relaxin 1 | −86 | **−2658** | −46 |
| IGKC | immunoglobulin kappa constant | −51 | −144 | **60** |
| IGHG1 | immunoglobulin heavy constant gamma 1 | −29 | −83 | **52** |
| SNORD3B-1 | small nucleolar RNA, C/D box 3B-1 | −3 | −35 | **49** |
| ACTG2 | actin, gamma 2, smooth muscle, enteric | −170 | −514 | **−561** |
| CNN1 | calponin 1, basic, smooth muscle | −403 | −680 | **−568** |
| NPY | neuropeptide Y | −50 | 348 | **−1345** |
| **Gene** | **Description** | **WIR-A** | **WIR-B** | **WIR-C** |
| IGLL5 | immunoglobulin lambda-like polypeptide 5 | **59** | 141 | 56 |
| TPM2 | tropomyosin 2 (beta) | **50** | 9 | −9 |
| ACTG2 | actin, gamma 2, smooth muscle, enteric | **46** | 2 | −29 |
| RPS8 | ribosomal protein S8 | **−1139** | −575 | −190 |
| RPL14 | ribosomal protein L14 | **−1182** | −734 | −85 |
| ZNF865 | zinc finger protein 865 | **−2073** | −1027 | −239 |
| IGLL5 | immunoglobulin lambda-like polypeptide 5 | 59 | **141** | 56 |
| MDK | midkine (neurite growth-promoting factor 2) | 28 | **133** | 148 |
| RPS27 | ribosomal protein S27 | −51 | **127** | 56 |
| NPIPB5 | nuclear pore complex interacting protein family, member B5 | −937 | **−1077** | −98 |
| KLK3 | kallikrein-related peptidase 3 | −585 | **−1248** | −439 |
| SPON2 | spondin 2, extracellular matrix protein | −332 | **−1575** | −1288 |
| MDK | midkine (neurite growth-promoting factor 2) | 28 | 133 | **148** |
| IFI27 | interferon, alpha-inducible protein 27 | 13 | 55 | **123** |
| HMGN2 | high mobility group nucleosomal binding domain 2 | 6 | 107 | **116** |
| CYTB | mitochondrially encoded cytochrome b | −238 | −446 | **−600** |
| LOC101929612 | mitochondrially encoded cytochrome c oxidase III | −97 | −279 | **−800** |
| SPON2 | spondin 2, extracellular matrix protein | −332 | −1575 | **−1288** |

### 3.5. Tumor Heterogeneity of Gene Networks

Figure 3 presents the expression coordinations of *AKT2* (v-akt murine thymoma viral oncogene homolog 2), with its partners central to the prostate-cancer development [57] in all four profiled regions from the patients "PQMZ" and "ABCN". Figure 3 also presents the coordinations of *MTOR* with mTORC1 (*RAPTOR*, ACT1S1, DEPTOR, MLST8, TELO2 and *TTI1*) and mTORC2 (*RICTOR*, MAPKAP1, PRR5, DEPTOR, MLST8, TELO2 and *TTI1*) partners [49] in the same regions.

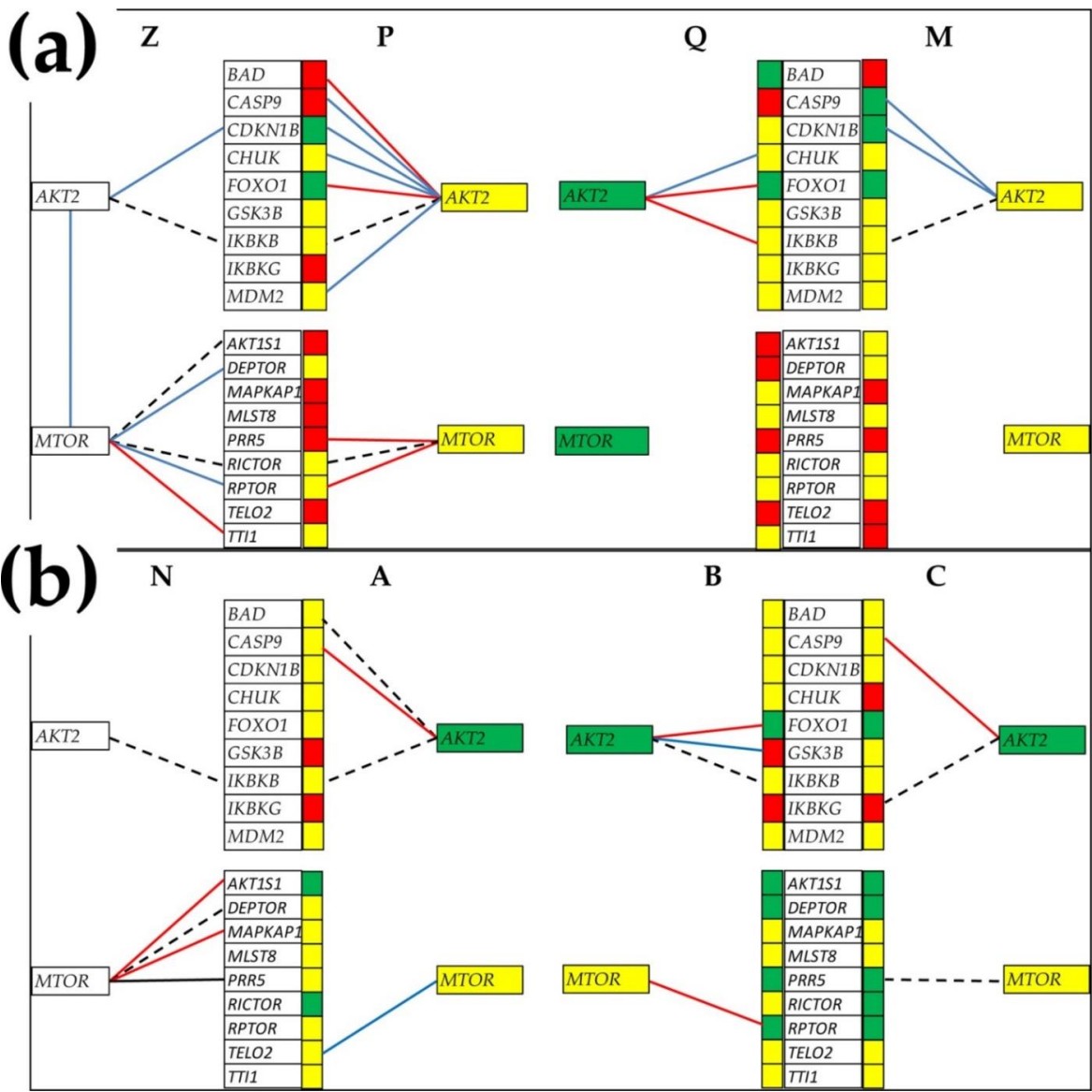

**Figure 3.** Expression coordinations of *AKT2* with its partners, central to the prostate cancer development, and the coordinations of *MTOR* with its partners from mTORC1 and mTORC2 in the four profiled regions of the (**a**) "PQMZ" patient and (**b**) the "ABCN" patient. A continuous red/blue line indicates a statistically (*p* < 0.05) significant synergistic/antagonistic expression of the linked gene, while a dashed black line indicates a statistically (*p* < 0.05) significant independent expression of the paired genes. Missing lines mean lack of statistical significance of the expression coordination between the two genes. Red/green background specifies significant up/downregulation of that gene in the indicated cancer nodule ("P", "Q", "M", "A", "B" and "C") with respect to the corresponding cancer-free surrounding tissue ("Z" or "N"), while yellow background means that the expression difference was not significant according to our composite criterion.

Of note, in Figure 3 are the substantial differences in both expression regulations with respect to the surrounding normal tissue and in expression coordination not only between the two patients but also among the cancer nodules of each patient. These results extend the notion of transcriptomic heterogeneity of the tumor [11–14] to the formation of gene networks that could be even more important for the cell behavior than the heterogeneity of the gene-expression levels.

Interestingly, *MTOR* is not significantly coordinately expressed with any of its MTORC1 and MTORC2 alleged partners in the nodules "Q", "M", "B" and "C", indicating major remodeling of the mTOR signaling in these cancer clones. Altogether, the differences in gene networking among the profiled groups of samples show that the pathways built by Kyoto Encyclopedia for Genes and Genomes (KEGG) [58] are not universal and can be used only as a general reference. The same conclusion is valid for the pathways built by other specialized software, such as Ingenuity Pathway Analysis [59], DAVID [60] and even the old GenMapp and MAPPFinder [61].

### 3.6. Gene Master Regulators

Figure 4 presents the GMRs of all the profiled regions from the two patients and the two cancer cell lines. For each GMR, the graph shows the GCH scores in all profiled sample types. Note that each group of samples has a distinct GMR and the substantial difference between the GCH score in the region the GMR commands and in the other regions from the same tumor. For instance, *FKBP9*, the GMR of the region "P" has the GCHs: 158.86 (in "P"), but 3.66 (in "Q"), 1.96 (in "M"), 2.59 (in "Z"), 8.24 (in "A"), 2.52 (in "B"), 2.36 (in "C"), 1.47 (in "N"), 7.46 (in "L") and 16.59 (in "D"). The large difference between the GCH scores in the cancer clone and the normal tissue indicates that manipulation of that gene would have major transcriptomic consequences in the cancer but practically nothing in the healthy tissue. This observation makes the GMR approach suitable to design cancer gene therapies.

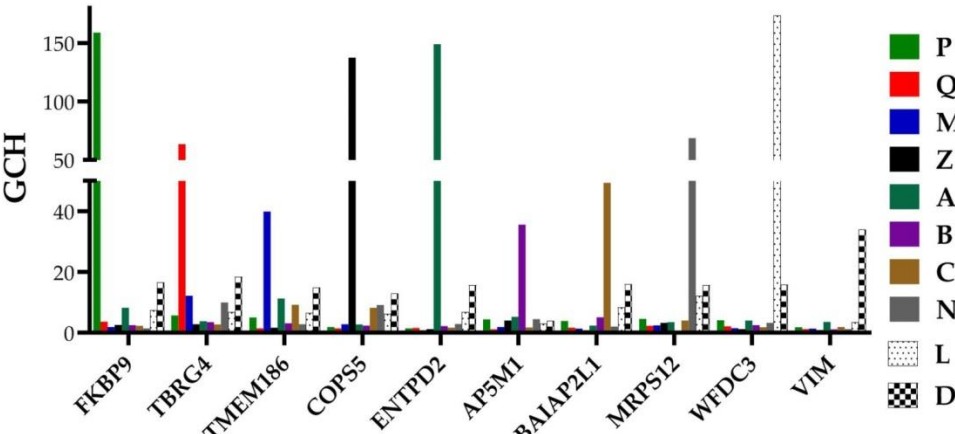

**Figure 4.** GMRs of the profiled regions from the prostates of the "PQMZ" and "ABCN" patients and from the cancer cell lines "L" and "D". Note that the GCH scores in the other samples are substantially lower the score in the sample commanded by the GMR.

The GMRs of the cancer nodules are not necessary regulated with respect to the normal tissue, but as evident from Table 2, they are among the most stably expressed genes in that region. The expression level of the GMR is allowed to fluctuate within a very narrow interval, because it regulates the expression of numerous other genes. For instance, *FKBP9*, the GMR of "P", was similarly downregulated in all three cancer nodules: x = −1.52 (WIR = −7.44) in "P", x = −1.59 (WIR = −8.51) in "Q" and x = −1.49 (WIR = −6.97) in "M". *TBRG4*, the GMR of "Q", was upregulated in "P" (x = 1.25, WIR = 0.46) and "Q" (x = 1.73, WIR = 1.39) but not in "M", while *TMEM186*, the GMR of "M", was not regulated in any of the profiled cancer nodules. *ENTPD2*, the GMR of "A", was not regulated in "A", but it was upregulated in "B" and "C"; *AP5M1*, the GMR of "B", was upregulated in "B" but not

regulated in either "A" or "C"; and *BAIAP2L1*, the GMR of "C", was upregulated in "B" and "C", but not in "A".

### 3.7. What Experimental Manipulation of the GMR Would Do to the Cancer Nodule's Metabolism?

We used our software #PATHWAY# [22] to identify all KEGG pathways [58] that include genes with statistically significant synergistic/antagonistic expression correlation with the GMR in each profiled cancer nodule. In all cancer nodules from both patients, the most significantly correlated genes with the GMR were from the metabolic pathways, indicating that targeting the GMR would have most dramatic consequences on the cell metabolism.

Figure 5 presents the significantly synergistically and antagonistically expressed genes with the corresponding GMR in each of the nodules "P", "Q" and "M" of the "PQMZ" patient, and what one might expect from the significant manipulation of the GMR expression. Thus, by therapeutically increasing the expression of *FKBP9* (downregulated in all three cancer nodules with respect to "Z"), the expressions of its synergistic partner genes would be pushed up, while the antagonistically expressed ones would be pushed down. Although the proposed therapeutic overexpression will restore the normal expression of *FKBP9* in all three cancer nodules, while upregulating it in "Z", it would have significant consequences only on "P", owing to the low *FKBP9* GCH scores in the other three regions. Large metabolic disturbances on the respective commanded cancer nodules, but not in the other regions, are also expected by knocking down *TBRG4* and *TMEM186*, as illustrated in panels Figure 5b,c.

Figure 6 presents the significantly synergistically and antagonistically expressed genes with the corresponding GMR in each of the nodules "A", "B" and "C" of the "ABCN" patient, and what one might expect from the significant manipulation of the GMR expression. Of note are, again, the different regulations of the metabolic genes in the cancer nodules of the same tumor, as well as between the two tumors (compare with Figure 5).

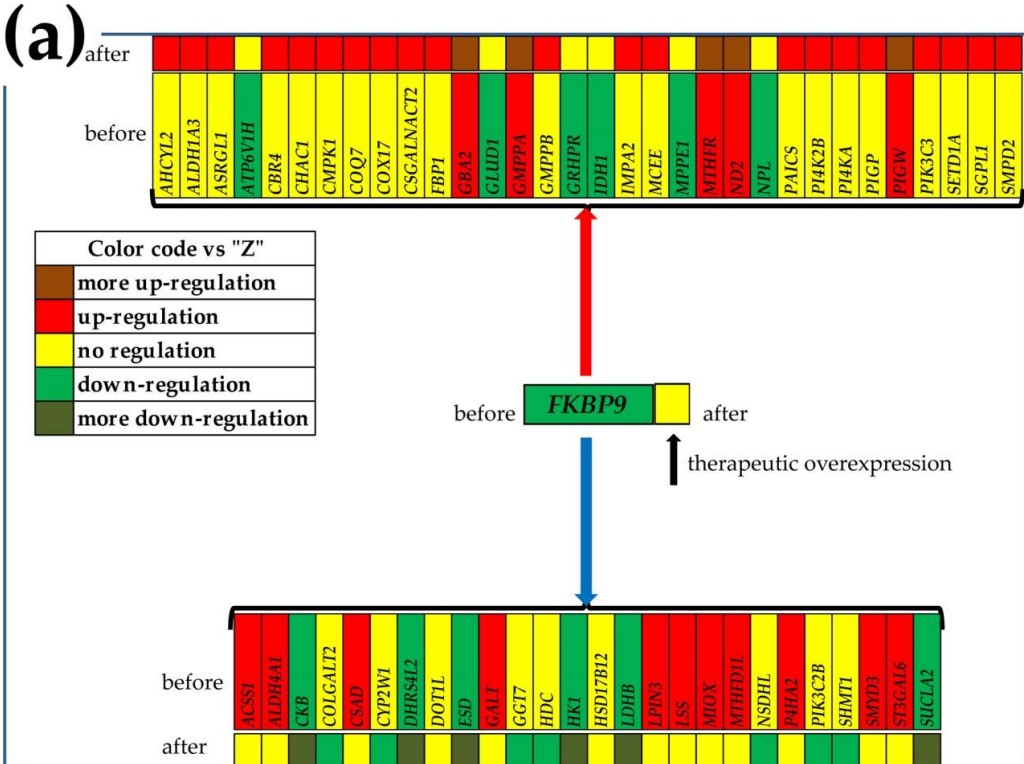

**Figure 5.** *Cont.*

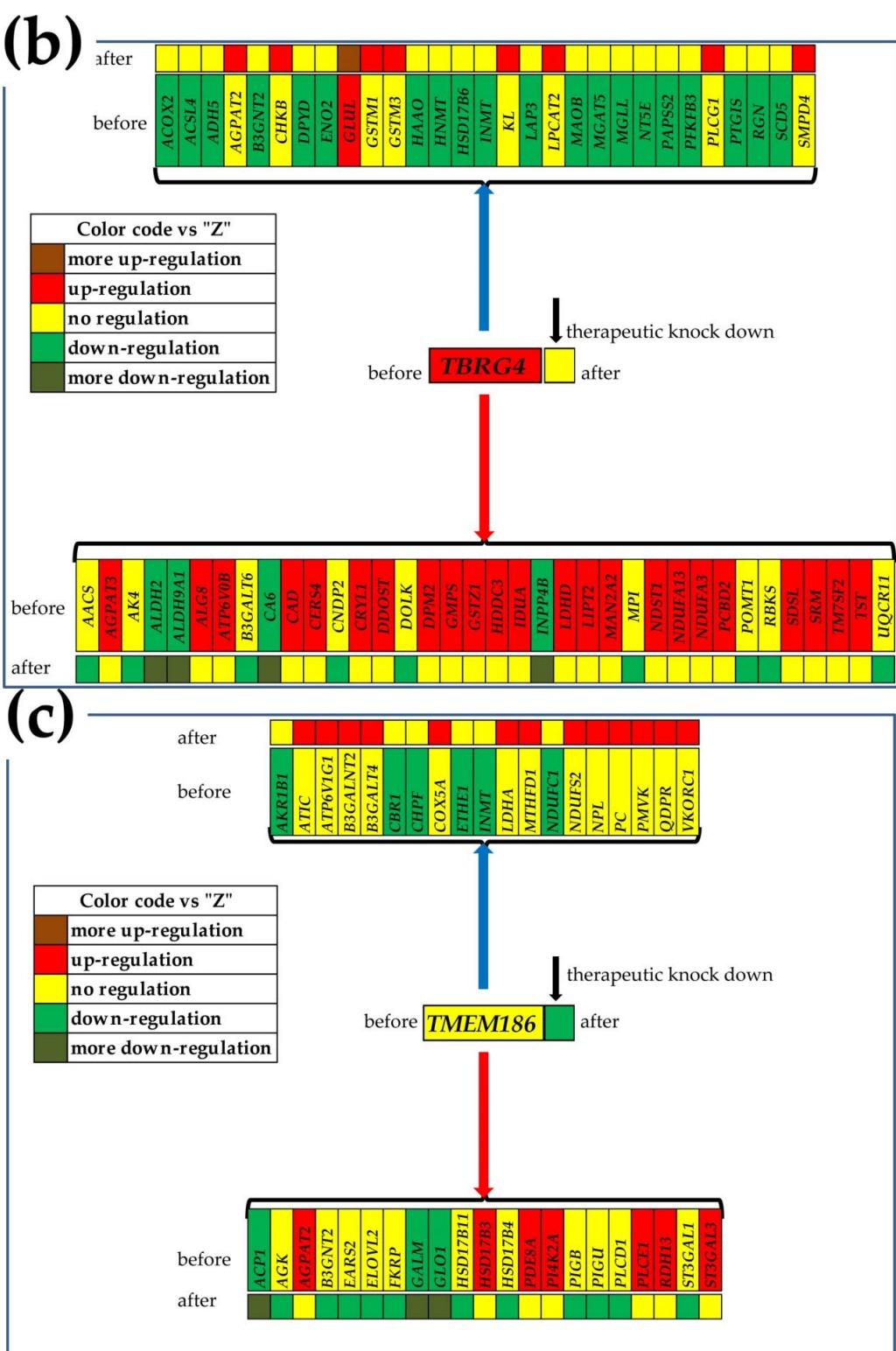

**Figure 5.** Significantly synergistically and antagonistically expressed metabolic genes with the corresponding GMR in the nodules (**a**) "P", (**b**) "Q" and (**c**) "M" of the "PQMZ" patient and the predicted regulations after the therapeutic alteration of the GMR. The red/blue arrow indicates the genes synergistically/antagonistically expressed with the GMR in that nodule. Gene symbol background indicates the status of that gene in the mentioned cancer nodule with respect to the surrounding normal tissue "Z" before (observed) and after the treatment (predicted). Note the different regulations of the metabolic genes in the cancer nodules.

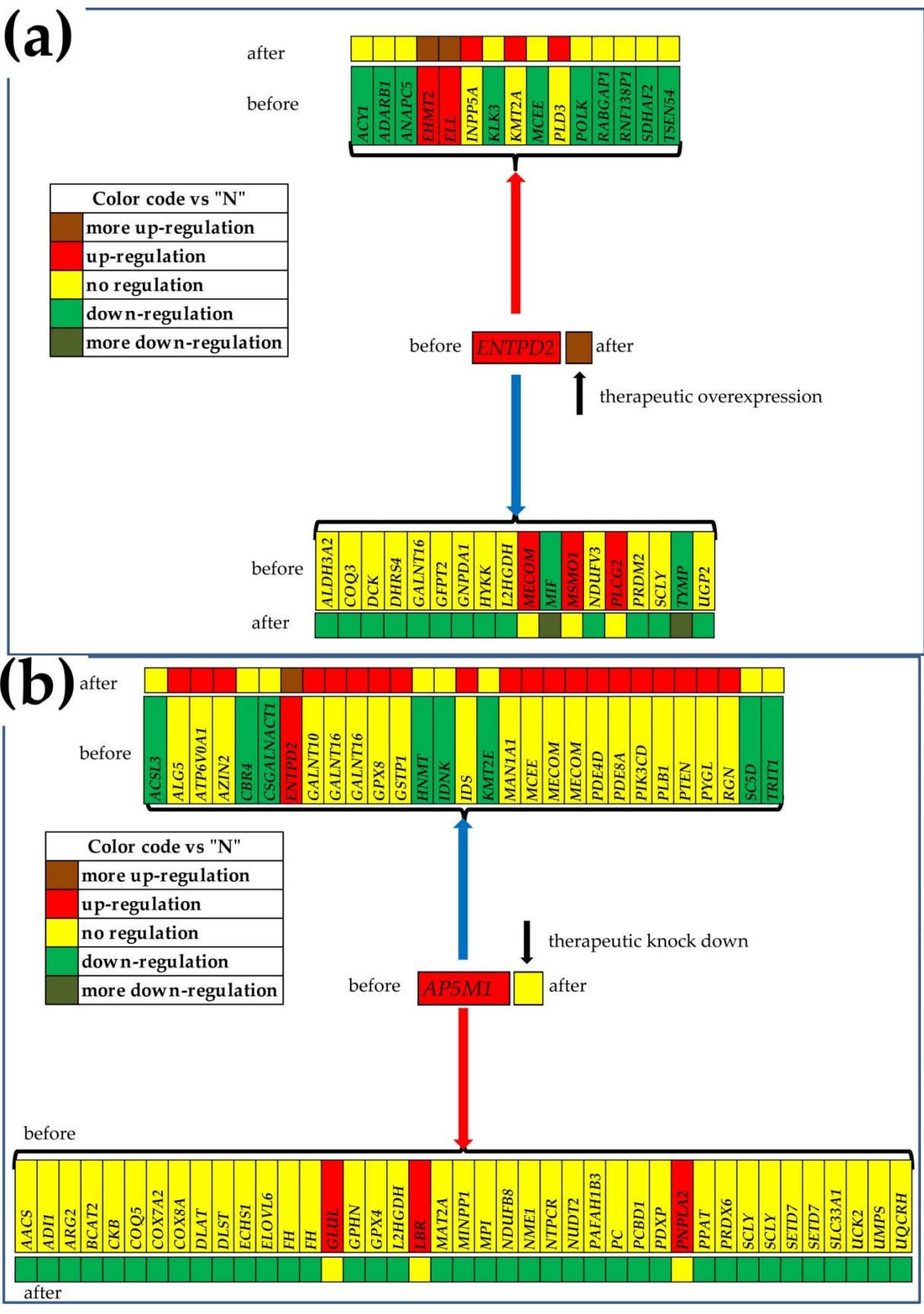

**Figure 6.** *Cont.*

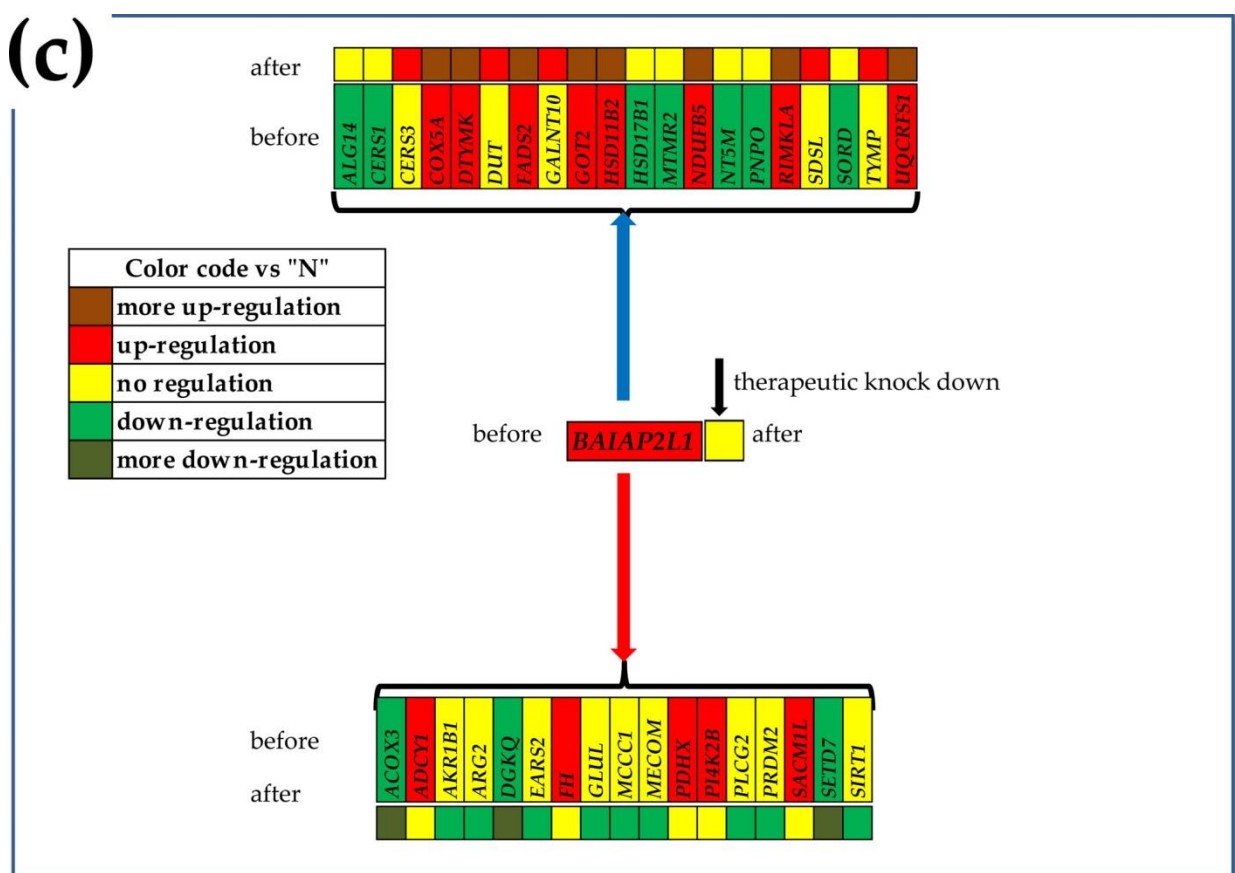

**Figure 6.** Significantly synergistically and antagonistically expressed metabolic genes with the corresponding GMR in the nodules (**a**) "A", (**b**) "B" and (**c**) "C" of the "ABCN" patient and the predicted regulations after the therapeutic alteration of the GMR. The red/blue arrow indicates the genes synergistically/antagonistically expressed with the GMR in that nodule. Gene symbol background indicates the status of that gene in the mentioned cancer nodule with respect to the surrounding normal tissue "N".

## 4. Discussion

The purpose of this study was to provide justification and a framework for the development of a personalized gene-therapy approach for prostate-cancer management. The manuscript detailed the theoretical foundations for a new and effective prostate cancer treatment. As defined by the US Food and Drug Administration (FDA), the gene therapy "seeks to modify or manipulate the expression of a gene or to alter the biological properties of living cells for therapeutic use" [62]. The FDA also issued the Guidance for Industry for "Long term follow-up after administration of human gene therapy products" [63]. According to the FDA [62], gene therapy replaces a disease-causing altered gene with a healthy copy of it or/and inactivates a disease-causing altered gene that is not functioning properly or/and introduces a new or modified gene. Gene-therapy products can be delivered as plasmid DNAs, using modified viral or bacterial vectors, or through gene editing technology. As such, gene therapy is part of the targeted therapies that, in the vision of the USA National Cancer Institute, include hormone therapies, signal transduction inhibitors, gene-expression modulators, apoptosis inducers, angiogenesis inhibitors, immunotherapies and toxin-delivery molecules [64].

The development of effective PCa therapies targeting selected genes or their downstream products was the objective of many research groups, and numerous publications detail their findings (e.g., References [65–69]. Some therapies that work against the consequences of altered genes (e.g., use of olaparib for HRR gene-mutated metastatic castration-resistant prostate cancer [70,71]) were already granted FDA approval (all FDA-approved

therapies for PCa are listed in Reference [72] and partially discussed in Reference [73]). Gene therapy targets not only protein-coding genes but also long non-coding RNAs [74] and microRNAs [75]. PCa FDA-approved therapies also include inhibitors of the andro-gen receptor signaling, such as enzalutamide [76,77] and darolutamide [78,79]. However, the endless diversity and the strong impact of certain individual characteristics on the PCa progression and treatment outcomes raise serious doubts about the value of such "good-for-everybody" targeted therapies, refocusing the research on personalized solutions.

The report is based on transcriptomic data obtained in IacobasLab from 10 groups of samples: three distinct cancer nodules and the surrounding normal tissue from each of two PCa patients and two standard PCa cell lines. All experimental data were documented in the NCBI Gene Expression Omnibus. Because we tailor our approach of the prostate-cancer genomics and gene therapy on the uniqueness of each affected man, and the data clearly show that, even within the same tumor, each cancer nodule has a distinct transcriptome, the sample size is not important.

The power of the Genomic Fabric Paradigm [18] used here comes from extending the workable transcriptomic information by four orders of magnitude by considering for each quantified gene all the readily available independent characteristics, namely AVE, REV and COR, with each other gene. While the AVE (the average expression level across biological replicas) is used in all studies to determine whether the gene was significantly regulated in cancer with respect to the normal tissue, the ignored REV and COR bring additional fundamental information. Thus, REV (relative expression variability) tells about a cell's priorities in limiting the expression fluctuations of that gene, with genes critical for the cellular phenotypic expression constrained within very narrow intervals (low REV). On the other hand, COR (expression correlation) shows how much the expression fluctuations of one gene are correlated with the fluctuations of another gene; such information is essential for the formation and maintenance of gene networks.

The independence and complementarity of these types of characteristics was proved in Figure 1 for the first 50 alphabetically ordered genes from the mTOR signaling pathway in the three cancer nodules and the surrounding normal prostate tissue. These genes were used only to illustrate the independence of AVE, REV and COR, but any other gene subset would equally prove the independence. It was also proved in previous publications for apoptosis in human thyroid [21], chemokine signaling in human kidney [18] and mouse cortex [80], evading apoptosis in human prostate [33], ionic channels in mouse heart [81] and PI3K–AKT signaling in mouse hippocampus [82]).

Our research confirmed the numerous reports about the heterogeneity of PCa tu-mors at the gene-expression-profile level [10–17]. For this, we included Table 1 for the genes with the largest expression in each profiled region, and Table 3 and Figure 2 for the genes' contributions to the cancer transcriptomic departure from the normal tissue. Use of the Weighted Individual Gene Regulation (WIR) and of the Weighted Pathway Regulation (WPR) provided more accurate measures of the transcriptomic alterations than the traditional percentages of upregulated and downregulated genes. Our results with distinct regulations even at the cancer nodule level within the same tumor (see Table 3 and Figure 2 in the present report and Figures 2–4 in [18] and Figures 2–4 in Reference [33]) question the idea of "transcriptomic signature" that is believed to be common for large cancer-affected populations. We consider such cancer transcriptomic signatures as artefacts of the meta-analyses combining transcriptomic data from many individuals (not always demographically grouped as race, age and other major criteria), collected by numerous labs, using (some)times distinct transcriptomic platforms and experimental protocols.

More importantly, we proved that the heterogeneity extends to the expression control (see Table 2 for the most stably and Supplementary Table S1 for the most unstably expressed genes). It extends also to the expression coordination (see Figure 3 for the networking of *AKT2* and *MTOR* with their partners). These heterogeneities are even more important than the heterogeneity of the expression level, because manipulation of the same gene may have different consequences in the distinct regions of the tumor. Thus, experimental overex-

pression or knockdown of one gene may trigger different responses of the homeostatic control mechanisms (higher in regions with low REV) and may differently remodel the gene networks.

The substantial heterogeneity of the gene-expression level and control and networking other genes require us to go beyond the bulk tissue and profile each cancer clone in the tumor separately. This strategy (when served by adequate analytical tools) is a pre-requisite for designing personalized therapeutic strategies to selectively destroy the cancer clones with minimal impact on the surrounding healthy tissue.

We have introduced the GCH (Gene Commanding Height) score to determine how influential a gene is in a particular region/condition and identified the GMR (Gene Master Regulator) of each profiled phenotype as its top gene (highest GCH). The GMR enjoys the strongest protection of expression (lowest REV), while being coordinately expressed with many other genes. As such, "smart" manipulation of the GMR expression beyond critical limits is expected to selectively kill (or at least block the proliferation) of the cells it commands. Since the GMRs of the cancer clones have very low GCH in the surrounding healthy tissue, targeting the cancer GMRs would have little effect on the normal cells. However, these are theoretical predictions that need further experimental validation.

Many investigators looked before us for transcription factors (e.g., References [83,84]) or hormone receptors (e.g., Reference [85]) as master regulators that can be used in targeted cancer therapies. However, our GMR approach is fundamentally distinct from the traditional quest for master regulators both by the selection method and by the gene coverage. Thus, instead of the molecular affinity (same regardless the cell phenotype) used to identify the transcription factors (see References [86–88]), we rank the genes according to their expression control and coordination with other genes (GCH) in that particular specimen. The GCH-based gene hierarchy is not only personalized for the profiled tissue, but it is also dynamic, being reorganized during the progression of the disease, in response to a treatment and to other environmental changes. Moreover, we do not restrict the GMR quest to transcription factors and hormone receptors; any coding and non-coding RNA can be chosen by the cell itself as its GMR if its strictly controlled expression level regulates major functional pathways via expression coordination with their genes.

Figure 4 shows that the GMR in one region has practically very little role in any other regions (much lower GCH), which is exceptionally important for designing a gene therapy that would selectively affect the phenotype ruled by the GMR.

In this report for metabolism (Figures 5 and 6) and in previous publications for basal transcription, RNA polymerase and cell cycle [18], apoptosis [21] and enzymes [33], we included predictions of what might happen when targeting the GMR by a gene therapy. Unfortunately, there was no possibility to experimentally validate any of these predictions on the patients from which we have collected the prostate tissues. However, the GMR approach involving a monotonically ascendant relationship between the GCH score and the overall transcriptomic changes was validated by us on two human thyroid cancer cell lines stably transfected with four genes [20,22]. Thus, transfection of the BCPAP (papillary) and 850C (anaplastic) thyroid cancer cells with *DDX19B* (DEAD-Box Helicase 19B), *NEMP1* (nuclear envelope integral membrane protein 1), *PANK2* (pantothenate kinase 2) and *UBALD1* (UBA-like domain containing 1) induced significantly larger transcriptomic alterations in the cells where these genes had higher GCH.

The worst scenario of our GMR approach is when manipulation of a gene with top GCH scores in two cancer nodules is beneficial for one cancer nodule but detrimental for the other. Although this situation is very unlikely (never found something even close in our studies), the solution is to go to the next in line gene that has either similar effects in the two nodules or is irrelevant (low GCH) in the second nodule.

## 5. Conclusions

For now, the approved FDA PCa treatments [72] are considered effective for all men, regardless of race, age, medical history, habits and other risk factors whose dynamic

combination makes each of them unique at each stage. However, there is enough evidence that each man responds differently to the same treatment, that the outcomes change in time for the same person and that, in most cases, there is little improvement. In contrast, we propose to identify the most legitimate gene targets that will selectively destroy the cancer clones of the prostate of the current patient, now. Although a "good-for-everybody" drug seems much more advantageous from economical point of view, in time, our approach may become economically competitive. There are already numerous FDA-approved cellular and gene therapy products [89], and, with the right stimulus, the industry will soon produce drugs based on CRISPR or other types of constructs to target almost all genes. When this will be the case, the oncologist will perform the transcriptomic analysis of tumor biopsies, identify the GMR of the cancer clones, and order and administrate the right product to his patient. The treatment will have similar costs as the actual gene therapy but would be more efficient for the PCa-affected person. Certainly, there are many more men with PCa (>3.1 mil diagnosed in the USA [90]) than protein-coding genes (~20,000), so that, on average, 15 of them share the same protein-coding GMR in one cancer nodule. However, we recommend a personalized three-gene cocktail to destroy the most aggressive three cancer clones in the prostate, and the chance of two men sharing the same gene triplet is less than 1 in one trillion.

Caution: Although very attractive, the GMR approach of the PCa therapy is still a theory that needs rigorous experimental validation.

If the tumor heterogeneity is very high (as determined by scRNA-sequencing), then one may have too many GMRs to consider. In such a case, one may use our method to establish the gene hierarchy for the entire tumor, considering the average GCH scores across the distinct nodules. The same could be performed by re-analyzing the gene expression profiles of many persons from the GFP perspective. Although not lethal for any nodule, the top genes will have enough influence on all of them. Apparently, this method will restore the concept of "one gene fits all", but with a totally different way to select the target gene(s).

**Supplementary Materials:** The following supporting information can be downloaded at https://www.mdpi.com/article/10.3390/cimb44010027/s1. Table S1: Three genes with the largest expression variability across biological replicas in all profiled regions of the two patients.

**Author Contributions:** Conceptualization, D.A.I.; methodology, S.I. and D.A.I.; software, D.A.I.; validation, S.I. and D.A.I.; formal analysis, D.A.I.; investigation, D.A.I.; resources, D.A.I.; data curation, D.A.I.; writing—original draft preparation, S.I.; writing—review and editing, D.A.I.; visualization, D.A.I.; supervision, D.A.I.; project administration, D.A.I.; funding acquisition, D.A.I. All authors have read and agreed to the published version of the manuscript.

**Funding:** This research received no external funding.

**Institutional Review Board Statement:** The study was conducted according to the guidelines of the Declaration of Helsinki. At the time of the experiment (2016), the study was part of D.A. Iacobas' project approved by the Institutional Review Boards (IRB) of the New York Medical College's (NYMC) and Westchester Medical Center (WMC) Committees for Protection of Human Subjects. The approved IRB (L11,376 from 2 October 2015) granted access to frozen cancer specimens from the WMC Pathology Archives and depersonalized pathology reports, waiving the patient's informed consent.

**Informed Consent Statement:** Patient consent was waived due to the use of depersonalized pathology reports.

**Data Availability Statement:** Raw and processed gene-expression data were deposited and are publicly accessible at https://www.ncbi.nlm.nih.gov/geo/query/acc.cgi?acc=GSE72333, https://www.ncbi.nlm.nih.gov/geo/query/acc.cgi?acc=GSE72414, accessed on 1 September 2021. https://www.ncbi.nlm.nih.gov/geo/query/acc.cgi?acc=GSE133891, accessed on 1 September 2021. https://www.ncbi.nlm.nih.gov/geo/query/acc.cgi?&acc=GSE133906, accessed on 1 September 2021. https://www.ncbi.nlm.nih.gov/geo/query/acc.cgi?acc=GSE168718, accessed on 1 September 2021. https://www.ncbi.nlm.nih.gov/geo/query/acc.cgi?acc=GSE183889. accessed on 1 September 2021.

**Acknowledgments:** D.A.I. was funded by the Texas A&M University System Chancellor's Research Initiative (CRI) for the Center for Computational Systems Biology at Prairie View University.

**Conflicts of Interest:** The authors declare no conflict of interest.

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
