# Peer review of "Personalized 3-Gene Panel for Prostate Cancer Target Therapy"

_cimb, doi:10.3390/cimb44010027_

Round 1

Reviewer 1 Report

I BELIEVE THAT THE PAPER COULD BE PUBLISHED IN THE PRESENT FORM. IT PRESENTS IN DETAIL A TOPIC OF GREAT CLINICAL INTEREST, PROVIDING THE FOUNDATIONS FOR A NEW AND EFFECTIVE PROSTATE CANCER TREATMENT.

Author Response

Thank you so much for the very kind appreciation of our work.

Reviewer 2 Report

In the research paper entitled “Personalized 3-gene panel for prostate cancer target therapy”, the authors proposed a Genomic Fabric Paradigm to identify the master regulators of cancer cells from gene expression data and predict the consequences of their experimental manipulation. The authors used microarray data for their analysis and similarly pointed out that any data can be used to achieve the outcome. Overall, the data is quite interesting, but the data representation can be improvised. The authors used quite confusing sentences which makes it difficult for a reader to understand. Please reframe the sentences which are long and correct grammar in a few paragraphs. The authors should also select a master regulator gene which they find absolutely important for the two cell lines (LNCaP and DU145) and perform knockdown experiments to validate their observations. It gives functional significance to their work. They can perform simple experiments to assess proliferation and apoptosis after the knockdown. If you cannot perform them, please give a reason why?

Author Response

Overall, the data is quite interesting, but the data representation can be improvised.
We agree that “data representation can be improved” and did our best, especially in the tables.
The authors used quite confusing sentences which makes it difficult for a reader to understand. Please reframe the sentences which are long and correct grammar in a few paragraphs.
We rephrased all long sentences and corrected the grammar.
The authors should also select a master regulator gene which they find absolutely important for the two cell lines (LNCaP and DU145) and perform knockdown experiments to validate their observations. It gives functional significance to their work. They can perform simple experiments to assess proliferation and apoptosis after the knockdown. If you cannot perform them, please give a reason why?
We recognize the importance of such an experiment and have the expertise and the experience to do it. However, with one of us (SI) retired and the other (DAI) moving his lab in another building, it will take months until we will have the needed resources on place. Nevertheless, we proved previously [20-22] beyond statistical doubt by stable transfection of DDX19B, NEMP1, PANJK2 and UBALD1 on the papillary BPAP and anaplastic 8505C human thyroid cancer cell lines that the transcriptomic effects are significantly larger on the cells where the manipulated gene has higher GCH score.

Reviewer 3 Report

The article under review uses an interesting bioinformatics approach to analyze  gene expression data from different nodules on the same prostate tumor, as well as in two different patients. Through their analysis, they come up with what is basically a ranked list of genes to be considered as master regulators for that particular tumor nodule. They suggest that each tumor, or each nodule will be different, and knowing the master regulator will enable finely specific treatment.

The article is largely theoretical in nature.

The authors published what appears to be a similar article in the journal Cells earlier in 2021. Can the authors discuss how the current work differs and is an advancement?

These studies utilize prostate cancer tumor samples from two patients and they are described as a 65 year old black man and a 47 year old white man. This race description is carried forward in line 217 where one sample is referred to as from the "white patient". There is no scientific reason to identify the race in these two samples. It is irrelevant to the study. This study has nothing to do with racial disparities. Being there is no biological basis for race, the identification of race in this study gives the impression that it does indeed matter biologically. Reference to the race of the samples should be removed entirely. In my opinion, the tumors should simply be identified as from "patient A and patient B", or "patient ABCN and patient PQMZ".

In paragraph 3 of page 8, the authors say that EIF4E works as an activator of MTOR in one sample, but an inhibitor in the other sample. I do not think this can be said, based on that they are simply upregulated together, or expressed in opposite directions. To say one is an activator/inhibitor of the other would seem to require cell and molecular biology experiments, which were not done in this study. The relationship between these genes should be described differently.

On line 590 the authors say that there is no possibility to validate experimentally any of the predictions on the patient tissues, but then they devote the rest of the paragraph to a past study in which they did validate their predictions on cell lines. Would it also not be possible to validate the results in prostate cancer in prostate cancer cell lines? If there is no possibility of validation of these studies, how much confidence can I have in them? It would appear to come down to a matter of 'belief'. I do not think the authors intended this to be the conclusion one might get from the paragraph.

In the paragraph at line 601, they authors say "worst case scenario" is that among the multiple tumor nodules in a patient, a gene treatment that harms one nodule may benefit another, but they have never found anything close to that happening. These studies represent two patients with a handful of tumor nodules. It is a very small N from which to expect to see the scenario described. It is plausible that in the real world, a patient with many tumors could end up with parts that require opposing treatments. Although there are thousands of genes, I would predict the subset of genes that consistently pops to the surface during this analysis is much smaller. In fact, it is possible that it is a small, handful of genes. In a patient who has many tumor nodules, or many metastatic tumors, the diversity could be largely covered in a single individual. These are data we simply do not know and cannot infer from 2 patients.

The main argument of this paper is that the described analysis can be done as a source of knowledge to be used for designer gene therapies to treat each unique tumor. This will be superior to therapies that treat all tumors the same. The authors strongly suggest this will be superior. One could also argue as the heterogeneity of the  tumor burden in a patient increases, the greater number of master regulator genes one would need to target. At higher heterogeneity, existing therapies that are less specific may control the cancer better as they act more broadly. In other words, this study does suggest a different approach that may be better, but it may not be a "slam dunk" (my words).

Line 29, "hopped" should be "hoped".

Reviewer 4 Report

In this manuscript, Iacobas S and Iacobas DAA investigated a bioinformatic approach to identify “Gene master regulators” (GMR) of prostate cancer cells and predict therapeutic efficacy of their manipulation.

They analyzed microarray data on prostate cancer tissue samples derived from two patients with metastatic prostate cancer, comparing 3 nodes of cancer origin to 1 node of surrounding normal tissue.

Authors aimed to find the genes whose potential altered expression compared to normal tissue should have the largest consequences on cells’ metabolism.

They reported that each nodes of each patients displayed different genes expression and identified potential consequences of manipulation of that genes in the different nodes.

Authors concluded that the bioinformatic approach proposed here should be useful to identify a genes alteration profile shared by (at least) the majority of patients in order to design a real personalized therapy for prostate cancer.

Despite the basis of this work appear interesting, results showed here represent an umpteenth confirmation about the large heterogeneity of prostate cancer lesions.

No explanations are reported regarding the molecular mechanisms mentioned, the role of which on cancer survival remains theoretical.

Moreover some additional information would be needed.

  1. In Table 1, 2, 3 and Figure 1, 2 only results obtained on Patient#2 are shown. Why authors selected only this patient and why the other were excluded?
  2. Figure 4 shows the importance of several genes in cancer or normal prostate nodes, and in two cancer cell lines (LnCap and Du145). It is not clear what is the role of cell lines in this experiment. What information can be drawn from these evidence?
  3. Why authors are trying to apply their interesting approach on prostate cancer that is largely known as one of the most heterogeneous type of tumors?

Minor points:

-The hope of developing CRISPR constructs for all genes as an effective solution to treat cancer should be omitted in abstract.

-Several typos are present in the text.

Round 2

Reviewer 2 Report

The authors addressed all the points raised

Author Response

Thank you for reviewing our work

Reviewer 3 Report

Thank you for the revisions.

Author Response

Thank you for the useful critiques

Reviewer 4 Report

Authors must clearly highlight (in abstracts, discussion and conclusions) that these are preliminary evidence and that the conclusions remain theoretical in the absence of experimental data.
